# HFE Related Hemochromatosis: Uncovering the Inextricable Link between Iron Homeostasis and the Immunological System

**DOI:** 10.3390/ph12030122

**Published:** 2019-08-22

**Authors:** Graça Porto, Eugénia Cruz, Maria José Teles, Maria de Sousa

**Affiliations:** 1Hematology, Centro Hospitalar Universitário do Porto (CHUP), 4099-001 Porto, Portugal; 2Basic & Clinical Research on Iron Biology, Instituto de Investigação e Inovação em Saúde (I3S) & Instituto de Biologia Molecular e Celular (IBMC), 4200-135 Porto, Portugal; 3Molecular Pathology and Immunology, Instituto de Ciências Biomédicas Abel Salazar (ICBAS), Universidade do Porto, 4050-313 Porto, Portugal; 4Clinical Pathology, Centro Hospitalar Universitário de S. João (CHUSJ), 4200-319 Porto, Portugal

**Keywords:** hemochromatosis, HFE, natural history, T lymphocytes, MHC, CD8+ T cells, prevention

## Abstract

The *HFE* gene (OMIM 235200), most commonly associated with the genetic iron overload disorder Hemochromatosis, was identified by Feder et al. in 1996, as a major histocompatibilty complex (MHC) class I like gene, first designated human leukocyte antigen-H (HLA-H). This discovery was thus accomplished 20 years after the realization of the first link between the then “idiopathic” hemochromatosis and the human leukocyte antigens (HLA). The availability of a good genetic marker in subjects homozygous for the C282Y variant in HFE (hereditary Fe), the reliability in serum markers such as transferrin saturation and serum ferritin, plus the establishment of noninvasive methods for the estimation of hepatic iron overload, all transformed hemochromatosis into a unique age related disease where prevention became the major goal. We were challenged by the finding of iron overload in a 9-year-old boy homozygous for the C282Y HFE variant, with two brothers aged 11 and 5 also homozygous for the mutation. We report a 20 year follow-up during which the three boys were seen yearly with serial determinations of iron parameters and lymphocyte counts. This paper is divided in three sections: Learning, applying, and questioning. The result is the illustration of hemochromatosis as an age related disease in the transition from childhood to adult life and the confirmation of the inextricable link between iron overload and the cells of the immune system.

## 1. Introduction

The practice of vaccination and the use of antibiotics provoked a significant decrease of infectious disease in the spectrum of modern clinical practice. That decrease uncovered simultaneously the appearance of other diseases whose presentation relates mainly to age. HFE-related hemochromatosis is a genetic disorder of iron overload. The combination of some luck, progress in the immunology of transplantation, progress of the understanding of the regulation of iron metabolism, progress in genomics, progress in the development of transgenic mice and in the characterization of adaptive immunity cell populations resulted in a sequence of events responsible for permitting today the early diagnosis and prevention of the iron overload induced clinical manifestations of the disease. Human HFE-related hemochromatosis has been the subject of several reviews by prominent researchers in the field [1,2]. In general, emphasis in the pathophysiology of the disease has been focused on the role of hepcidin in iron homeostasis [3,4]. Less attention has been directed to the role of HFE in immunology [5,6,7] and to the potential contribution of lymphoid cell numbers to that same pathophysiology [8,9,10]. 

In the present paper, under “learning”, we briefly review the evidence sustaining the importance of immunology in HFE-related hemochromatosis, starting by the improbable discovery of its link with HLA to the discovery of the *HFE* gene as a MHC class I-like gene and its later association with selected immune defects. Under “applying”, we report a 20 year follow up of iron parameters and lymphocyte counts in three children homozygous for the C282Y HFE variant who started to express signs of iron overload in their transition from childhood to adult life. Under “questioning”, we explore the issue of the still unexplained immunological functions of HFE. The paper concludes with a brief overview of its implications for the clinical practice and for the immunological theory.

### Learning: A Brief Historical Sequence Where the Improbable Led to Discovery

In 1975 it was improbable to think that HLA could be of clinical importance beyond histocompatibility or diseases with an immunological background. However, Simon thought the improbable and encountered in Fauchet an immunogeneticist open to doing the HLA typing of 20 “idiopathic” hemochromatosis” patients to find 17 with HLA A3 [11]. The result was the discovery, first published in 1975 in French [11] and in 1976 in the Lancet paper with the title, “HLA and "non-immunological" disease: idiopathic haemochromatosis” [12]. 

The publications of Simon, Fauchet and coworkers were to provide some background to a postulate published in 1978 saying that the immunological system could have a role in the surveillance of iron toxicity [13]. That postulate was followed by a series of studies of immunological cell populations in patients with “idiopathic” hemochromatosis [8,9,14,15], preceding the finding by Feder et al. in 1996 of the hemochromatosis gene as a novel MHC class I-like gene [16], closing a circle of 21 years from the improbable to the discovery. 

Radical changes followed the discovery of the hemochromatosis gene and its association with the C282Y HFE variant. The introduction of genetic testing, combined with Magnetic Resonance Imaging (MRI) as a non-invasive measure of liver iron concentration, the systematic family screening, approaches to disease penetrance from large population studies, all led to a radical change in the clinical presentation and timing of diagnosis well demonstrated by the decreasing frequency of severe liver disease [17].

HFE-related hemochromatosis appears thus as a “dream like” age related disease, with a strong genetic marker, reliable and reproducible serum biochemical markers, namely transferrin saturation and serum ferritin, and confirmatory non-invasive tools all enabling diagnosis much before the clinical presentation of the disease.

The surprise, however, arises from the fact that a disease thought of as non-immunological, can be seen inevitably as an immunological disease. Firstly, the gene is an MHC class I like gene in strong linkage with other genes within the MHC cluster [18,19]. Secondly, the very early studies of lymphoid cell populations in patients with (at the time) idiopathic hemochromatosis demonstrated abnormally high CD4/CD8 ratios in those with a more severe iron overload [14]. At that time, it was also demonstrated that after complete iron depletion by repeated phlebotomies, entry of iron measured by changes in transferrin saturation was faster in patients with the highest CD4/CD8 ratios [14]. Later, a greater importance was attributed to the finding of low numbers of CD8+ T cells associated with the severity of iron overload [9,20] and the demonstration that the low CD8+ T lymphocyte numbers in hemochromatosis are due to defects in the most mature effector memory cells [21]. When those defects were described, they generated some surprise and confusion amongst researchers and clinicians because of the previously existing evidence of iron induced expansions of T lymphocyte populations, namely, relative expansions of CD8+ T lymphocytes, both in experimental models of Fe-citrate injection [22,23] and in clinical models of transfusional iron overload [24,25]. The reciprocal effect, i.e., that primary immune defects could, in turn, contribute to iron overload, was next confirmed with a number of experimental studies examining and confirming the presence of iron overload in mice deficient in selected [26,27,28] or total lymphocytes [29] and a more severe phenotype in mice lacking both HFE and β2-microglobulin [30]. Surprisingly, mice lacking only classical MHC class I molecules also developed iron overload [31]. 

The mechanisms underlying the lymphocyte defects in hemochromatosis are still poorly understood. The most recent evidence points to the possibility of a continuous effect where iron may sustain a constant activation, self-renewal and proliferation of CD8+ cells, and this may eventually lead to exhaustion of the effector memory T cells [32]. 

Regarding the mechanism “how” could lymphocyte defects contribute to iron overload, it is not until 2014 that Pinto and co-workers, in an extensive analysis of the interaction of lymphocytes with non-transferrin bound iron (NTBI), demonstrated that lymphocytes can take in NTBI in vitro [33] and that in vivo lymphocyte transfer can correct the iron overload of immune-deficient mice [10], closing thus another circle: From firm but unexplained observations to the demonstration of how lymphoid cell numbers could have a role in the control of iron overload, i.e., by acting as a circulating pool capable of “buffering” NTBI. 

Evidence of the inextricable connection between iron homeostasis and the adaptive immune system recently gained a novel impulse with the demonstration that patients with a homozygous p.Tyr20His mutation in the transferrin receptor 1 (TfR1) have a combined immunodeficiency characterized by normal numbers but impaired function of T and B cells [34]. Besides TfR1, other iron regulatory genes had also been previously found to be critical for lymphocyte activation and function, namely H-ferritin, whose conditional deletion in mice was shown to reduce B and T lymphocyte populations [35] or hepcidin, whose expression is increased during lymphocyte activation and shown to be necessary for proper lymphocyte proliferation [36].

With regard to the cells of the mononuclear/phagocytic system, their role in the recycling of senescent red blood cells provides perhaps the most significant illustration of the close interactions between iron metabolism and the immunological system [24]. In the case of secondary iron overload, as a result of dyserythropoiesis, hemolysis, or transfusions, macrophages are heavily loaded with iron which is released in the form of low-molecular weight (LMW) iron. This leads not only to increased transferrin saturation but also iron circulating as NTBI that will inappropriately enter tissues and cells [37]. In HFE-related hemochromatosis, the scenario is somehow different. In spite of the high transferrin saturation and circulating NTBI, as a result of increased iron absorption, little iron is seen in the Kupffer cells and other macrophages, while hepatocytes already show iron overload [2]. In a previous study of iron release by monocytes after erythrophagocytosis, Moura and co-workers demonstrated that monocytes from hemochromatosis patients released twice as much iron in a LMW form as control cells [38]. Based on those observations, they proposed for the first time the existence of a basic abnormality in the retention of iron in macrophages and probably from intestinal mucosal cells [38], a mechanism that is presently well established with the demonstration that hepcidin, which is functionally defective in hemochromatosis, regulates cellular iron efflux by binding to ferroportin and inducing its internalization [3]. Finally, one may question how the handling of iron in other compartments of the mononuclear/phagocytic system may also impact in their response to other types of toxicity. As an example, increasing evidence suggests that the accumulation of iron in the brain and the consequent microglia activation are hallmarks of neurodegenerative disorders, including Alzheimer’s disease [39]. However, substantial efforts are still needed to fully understand this and many other aspects of the complex interactions of iron with inflammation and immunity. In our view, HFE hemochromatosis continues to offer a particularly good model to approach new questions.

## 2. Results

### 2.1. Applying: A Case and Family Report

Armed with all the tools available in 1999 and the possibility to perform at our center systematic individual longitudinal studies, we were in a position to follow the real natural course of a disease, not just the course deduced from cross sectional data at different ages. Every clinician following a patient with an age related disease hopes to be able to have tools for an early detection of the disease, and fortunately, this is the case for hemochromatosis. We were challenged by the finding, by an attentive pediatrician, of a 9-year-old boy with an abnormally high transferrin saturation. This case, the subsequent family study with detection of two additional cases, and their clinical progression are described below. Highly motivated by the search of the reciprocal interactions between iron and the immunological system, we included in our clinical follow-up not only the iron related parameters but also the lymphocyte counts.

The proband was thus a 9-year-old boy, referred to our center for investigation of an abnormally high transferrin saturation (TS) value (65%) accidentally found on a routine examination and confirmed on a second determination (70%). He had a normal healthy growth and no signs or symptoms of any disease; the physical examination was normal and had normal blood counts for age. Serum ferritin (SF) was normal (79 ng/mL). *HFE* genotyping was performed with the parents’ informed consent and revealed homozygosity for the C282Y variant. Genetic counseling was offered to the family and consent obtained to perform *HFE* genotyping in the parents and in the two only brothers of 5 and 11 years of age, all apparently healthy. Results of the family screening are illustrated in Figure 1, showing that both mother and father were heterozygous for the C282Y variant, with normal iron parameters. 

Surprisingly, the two brothers were C282Y homozygous, with abnormally high TS (respectively 94% and 61%) with normal SF values of 41 ng/mL and 93 ng/mL, respectively. Of note, all 3 siblings had high lymphocyte counts as appropriate for their age. All procedures involving the collection of human samples and data were carried out following the rules of the Declaration of Helsinki of 1975, revised in 2013 [40], with the approval of the Ethical Committee of Centro Hospitalar Universitário do Porto.

The 3 siblings were regularly followed at our center with longitudinal determinations of iron parameters and lymphocyte counts for the last 20 years. They were recommended to become volunteer blood donors at the age of 18. Results of the follow-up are shown in the graphs on Figure 2 with the illustration of a representative case. 

The profiles of SF progression together with the change of lymphocyte counts offer a remarkable illustration of how early does iron start to accumulate in C282Y homozygous subjects, how it progresses during the transition from childhood to adult life and how the total numbers of lymphocytes may influence that progression. In all cases, SF started to be elevated before adult age (< 18 years) and continued to rise inversely to the fall of lymphocyte counts that continued to decline until reaching the adult levels at around 25 years (only case II-3 did not reach yet 25 years of age). The graphs also illustrate that intervention with intensive treatment was necessary to normalize SF in spite of regular blood donation. A better illustration of the inverse correlation between SF values and lymphocyte counts, before the impact of intensive treatment, is given in Figure 3 where the individual values in all 3 cases are plotted against age. 

Notably, case II-2, whose lymphocytes declined to values <1.5 × 10^6^/mL (considered lymphopenia) at the age of 26, reached SF levels much higher (>700 ng/mL) than his brother, case II-1, whom, at the same age, had lymphocytes stabilized at around 2.0 × 10^6^/mL, and a lower SF value (500 ng/mL).

In order to determine which sub-populations contributed the most for the setting of lymphocyte total numbers, we also determined, by flow cytometry analysis, the numbers of CD4+ T cells, CD8+ T cells and non-T subpopulations along the follow-up time. The results are shown in Figure 4 where it is clear that the change in lymphocyte counts is due to the continuous decline in T cell numbers and not related to the numbers of non-T cell populations (a combination of B and natural killer cells), which are more variable and probably related to immune responses against foreign antigens.

Naturally, the question could be raised if it is the lymphocyte depletion that is contributing to the progressive iron accumulation or, on the contrary, is the lymphocyte depletion a consequence of iron overload? In order to answer that question, we retrospectively reviewed the iron parameters and lymphocyte populations determined in 190 apparently healthy, non-C282Y homozygous family members of hemochromatosis patients, aged 5–20 years, who were studied in the context of a systematic family screening program at our center. The results, illustrated in Figure 5, confirm that the decline in lymphocyte counts is a normal physiological process and that, although SF slightly increases with age (particularly in males), it never reaches abnormal levels above 200 ng/mL in females or 300 ng/mL in males. These findings come to support the notion that, in order to start accumulating iron in hemochromatosis, it is necessary to combine two conditions: The genetic predisposition given by C282Y homozygosity to increase the iron in circulation and the lack of a sufficiently high number of T lymphocytes to buffer the excessive serum iron. 

### 2.2. Questioning: A Dual Function for HFE?

In spite of all advances supporting the cross-relationships between iron and adaptive immunity, many questions still remain unexplored and may keep immunologists and iron biologists busy for some time. Among them, perhaps the most critical question relates to the HFE function and how does it impact simultaneously on iron metabolism and on the adaptive immune functions. Interestingly, 20 years before the discovery of HFE, Svejgaard and Ryder postulated that HLA alloantigens could interfere with ligand/receptor interactions not directly involved in immune reactions, and those interactions could under certain conditions explain some associations between HLA and non-immunological diseases giving as an example hemochromatosis [41]. This view came to be vindicated with the demonstration that HFE, a non-classical HLA molecule, interacts with both the TfR1 to regulate transferrin-mediated iron uptake [42] and with transferrin receptor 2 (TfR2) as an iron sensor for hepcidin signaling [43]. The role of HFE, however, is not limited to iron metabolism; it is also implicated in adaptive immune functions. Sometime ago, it was suggested that HFE could be immunogenic [44], and more recently Reuben and co-workers proposed that HFE could have a role in antigen processing and presentation leading to an inhibition of CD8+ T-lymphocyte activation [7]. These were in vitro studies based on several T-lymphocyte activation read-outs in cells transfected with wild type and mutated HFE molecules. The first demonstration in vivo that HFE acts as a negative regulator of CD8+ T-lymphocyte activation and differentiation was provided with the study by Costa et al. [32] of lymphocyte gene expression signatures from HFE-related hemochromatosis patients and mouse models, where it was shown that the lack of HFE impacts on several activation markers in CD8+ T lymphocytes [32]. Whatever the mechanism how this interaction may happen, it may imply the cross-talk between HFE and the MHC class I antigen presentation pathway, as suggested by Almeida et al in their study of peripheral blood mononuclear cells from patients homozygous for the p.C282Y variant in HFE [5]. In that study they found a reduced cell-surface expression of MHC class I due to an enhanced endocytosis rate of MHC class I molecules caused by premature peptide and β2-microglobulin dissociation [5]. Whether this could happen through a direct effect or through the indirect influence of an unfolded protein response [45], this remains a pending question.

In summary, the mutated HFE can be now appropriately defined with a Janus like nature, implicated in two relevant pathways: Regulation of systemic iron homeostasis, through its interaction with the transferrin receptors and hepcidin signaling [42,43]; and the core of the immune system, by influencing MHC assembly and surface expression [5], and CD8+ lymphocyte expansion [7,32]. Failure of this dual function may explain the phenotypes observed in hemochromatosis of both iron overload and defective numbers of CD8+T lymphocytes. To better illustrate this view, we present in Figure 6a diagram summarizing how iron is essential for lymphocyte functions and, in turn, how lymphocytes, equipped with the capacity to uptake, hold and mobilize iron, contributing to the control of systemic iron homeostasis and the protection against iron toxicity.

At the systemic level, HFE is one of the players of the complex regulatory machinery of iron sensing and hepcidin signaling [43], thus contributing to the prevention of inappropriate iron accumulation. In HFE-related hemochromatosis, this regulation fails and the consequence is systemic iron overload. One of the multiple physiological implications of a well-controlled systemic iron homeostasis is the maintenance of appropriate adaptive immune functions. Lymphocyte responses depend on the interplay between the MHC dependent activation pathway, and their capacity to proliferate and differentiate, for which iron availability is essential through TfR1 mediated endocytosis [34]. The control of ferroportin mediated iron export through autocrine hepcidin signaling also impacts on the lymphocyte proliferation capacity of relevance [36], the hemochromatosis protein HFE was shown to have an additional role in the assembly and expression of the classical MHC class I for antigen presentation [5]. In the event of acquired systemic iron overload, such as occurs in transfusion dependent thalassemia patients or in experimental models with iron injection, lymphocytes (particularly CD8+ T lymphocytes) will be activated and expand in response to iron [22]. The recent demonstration that lymphocytes have the capacity to uptake the non-transferrin bound iron (NTBI) [33] and by doing so, to act as “buffers” of the systemic iron overload and protect from the iron accumulation in other tissues [10], came to vindicate the postulated reciprocal role of lymphocytes in the protection against iron toxicity [13]. Of relevance, the hemochromatosis protein HFE also has a role in this setting, by influencing the CD8+ T lymphocyte gene expression of molecules involved in activation and differentiation that finally have an impact on the number of circulating CD8+ T cells and consequently [32], on its NTBI “buffering” capacity. In summary, iron is essential for lymphocyte functions and, in turn, lymphocytes equipped with the capacity to uptake, hold, and mobilize iron contribute to the control of systemic iron homeostasis and the protection against iron toxicity. Pharmaceutical and/or therapeutic targets in the context of this new awareness are not completely obvious but evidence consistently points to TFR1 [34] and HFE as key molecules in the regulation of the crosstalk between the two systems. The molecule responsible for NTBI uptake by lymphocytes remains to be discovered. In summary, as usual, answers can only be found when the importance of the questions is acknowledged, and that takes time. The authors hope that the openness of Pharmaceuticals to the views presented here is a sign that the distance between questions and answers is being shortened.

## 3. Discussion

### 3.1. Implications for Disease Prevention in HFE-Related Hemochromatosis

The prevailing view regarding the onset of HFE-related hemochromatosis is that patients develop symptoms only at around 30–40 years of age [2]; it is well established, however, that they may have biochemical evidence of iron overload some time before [46]. Although early diagnosis is well accepted as the best certainty of therapy in this age related disease, no available guidelines specifically address the question of when should biochemical screening be performed on a population basis. Two large population-based prospective studies with long-term follow-up of untreated p.C282Y homozygous subjects described the progression of iron overload with age but including only subjects older than 35 years in one study [47] and older than 40 years in another [48]. In another HH cohort study with a long-term (24 years) follow-up of subjects screened by family or primary medical practice screening, it was estimated that the mean age of homozygotes who could develop a severe liver iron overload would be approximately 21 years after the hepatic iron stores begin to increase in both men and women [49]. In spite of this knowledge, the average age of diagnosis of HH patients continues to be at around 40 years [17], and there is still no evidence-based recommendation for screening earlier. The results described above challenge this view by showing that markers of the disease may be found much before adulthood and that, at least in some cases, at the age of 28 years there may be already a significant amount of iron accumulated. Knowing the silent toxicity of iron, the question of when should screening be ideally performed deserves a wider discussion.

### 3.2. Implications for Immunological Theory

The consistent associations found between iron overload and lymphocyte numbers or function vindicate the postulated notion that the immunological system has a surveillance role of the toxicity of iron [13] much before iron becomes a threat posed by its use by microbes [50]. How exactly that role is exercised is still not fully understood. Searching first in the β2-microglobulin (β2m) knock-out mice, spontaneous iron overload was found in organs such as liver and pancreas [26]. The subsequent finding of spontaneous hepatic iron overload in mice lacking classical MHC Class I molecules was surprising and signals the involvement of MHC class I itself in the exercise of control of iron overload [31]. A similar explanation may apply to the finding in the two separate studies by Rodrigues et al. [51] and Levy et al. [30] of higher hepatic iron overload in older β2m deficient mice than in HFE deficient mice. No matter how results affirm themselves, MHC class I appears always as a “culprit” or a motive for tissue iron accumulation: Either by its absence, such as in β2m knock out mice, or because its assembly and expression are affected by the C282Y mutated HFE. 

## Figures and Tables

**Figure 1 pharmaceuticals-12-00122-f001:**
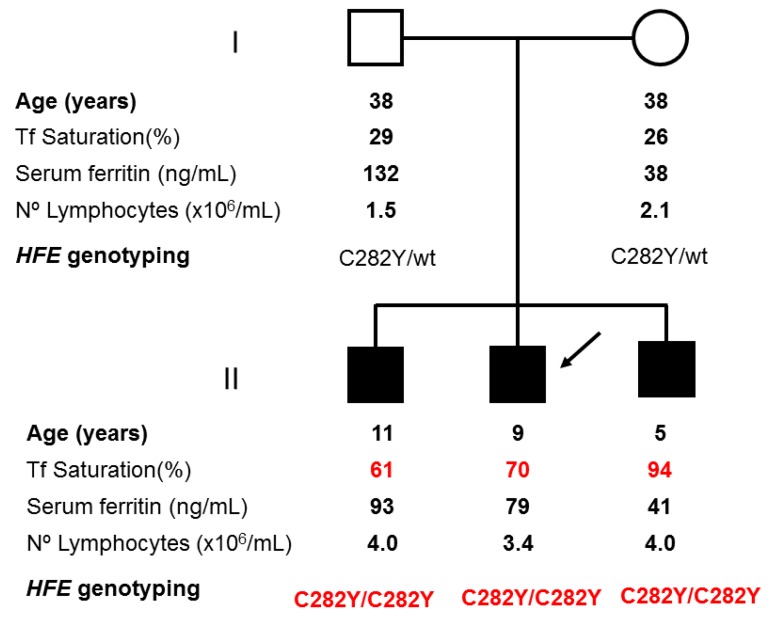
Family Pedigree of the case study. Age, serum markers of iron metabolism, total lymphocyte counts, and *HFE* (OMIM 235200) genotype are shown for each subject. The C282Y variant in the *HFE* gene is characteristically found in the homozygous form in patients with hemochromatosis. Homozygous subjects (C282Y/C282Y) are here represented by solid symbols and heterozygous (C282Y/wt) for half solid symbols. The proband is indicated by an arrow. Values in red are outside normal range.

**Figure 2 pharmaceuticals-12-00122-f002:**
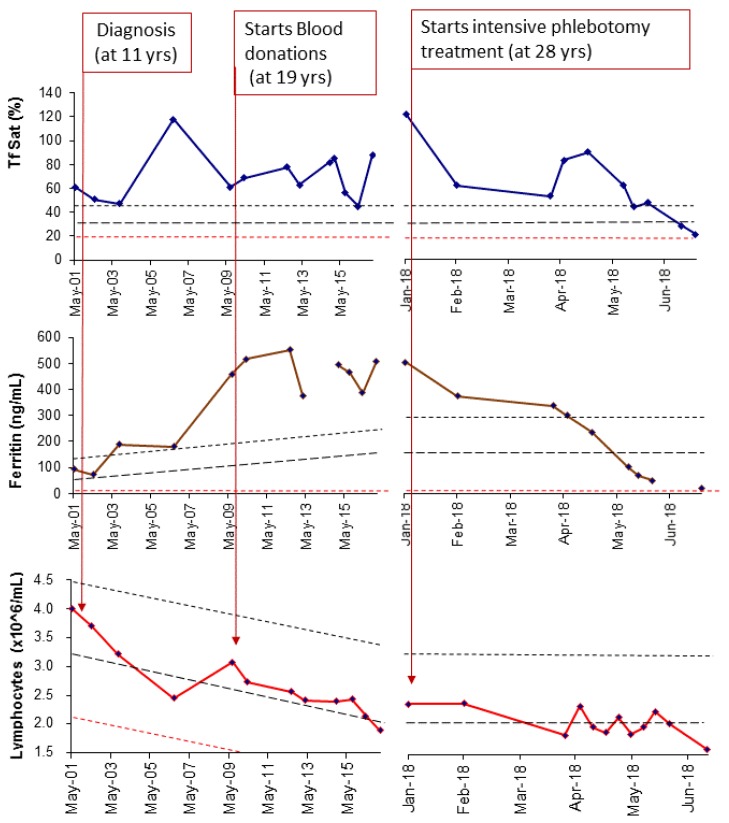
Follow-up of transferrin saturation (TfSat), serum ferritin, and total lymphocyte counts during a 17-year period. A representative case is shown (case II-1). Time at diagnosis, start of blood donation, and beginning of intensive phlebotomy treatment are marked in graphs by arrows. The large dashed lines represent the expected average values depending on age, derived from the simple regression equations in a population of sex and age matched normal controls recruited in the context of family studies. The small dashed lines represent the upper (in blue) and lower (in red) 95% confidence limits for each regression.

**Figure 3 pharmaceuticals-12-00122-f003:**
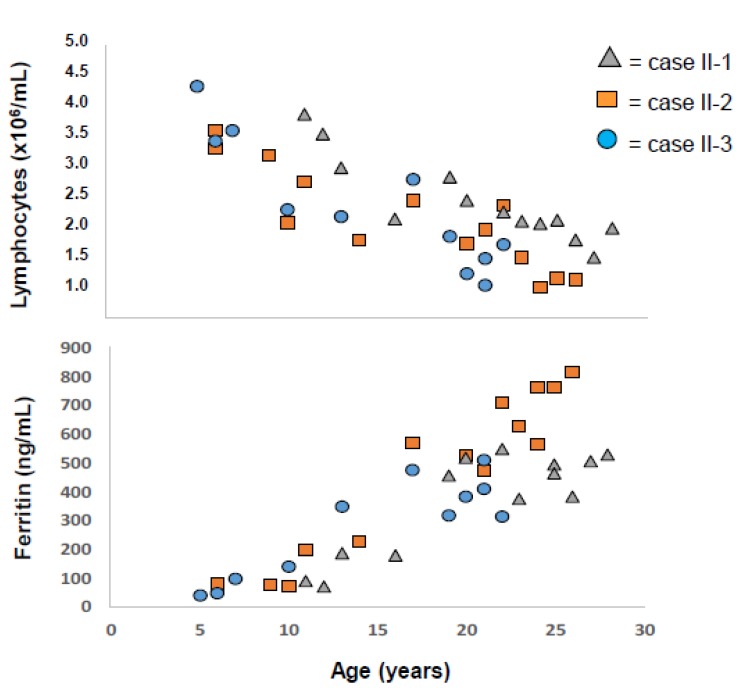
Inverse correlations between lymphocyte counts (**A**) and serum ferritin (**B**) before intensive treatment. Individual values from all three cases (II-1, II-2 and II-3) are plotted against age.

**Figure 4 pharmaceuticals-12-00122-f004:**
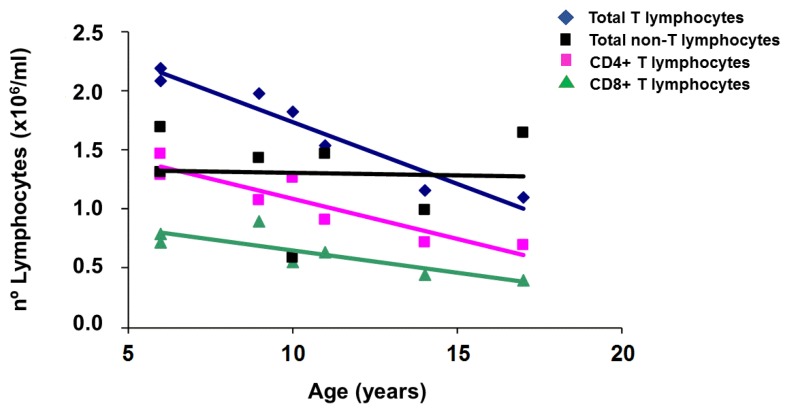
Follow-up of lymphocyte subpopulations. Total T-lymphocytes, CD4 and CD8 T lymphocytes, and non-T lymphocyte counts from a representative case (II-2) are plotted against age. Total T lymphocytes were calculated by the sum of CD4+ and CD8+ T cells. Non-T lymphocytes were calculated from the difference between the absolute lymphocyte numbers and the total T lymphocytes.

**Figure 5 pharmaceuticals-12-00122-f005:**
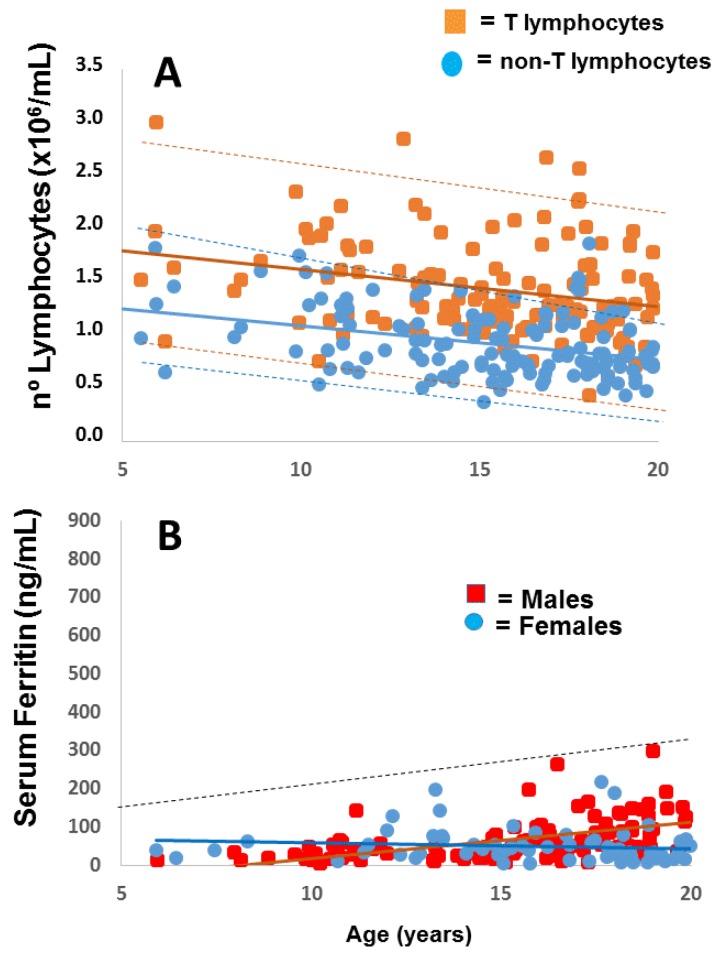
(**A**) Change in total T and non-T lymphocytes with age in non-homozygous apparently healthy family members screened before 20 years of age (*n* = 190). (**B**) Variation in serum ferritin with age in the same subjects, grouped as males or females. Full lines represent the regression lines for each correlation. Dashed lines in panel A represent the upper and lower 95% confidence limits for T (in brown) and non-T (in blue) lymphocytes. The dashed line in panel B represents the upper 95% limits.

**Figure 6 pharmaceuticals-12-00122-f006:**
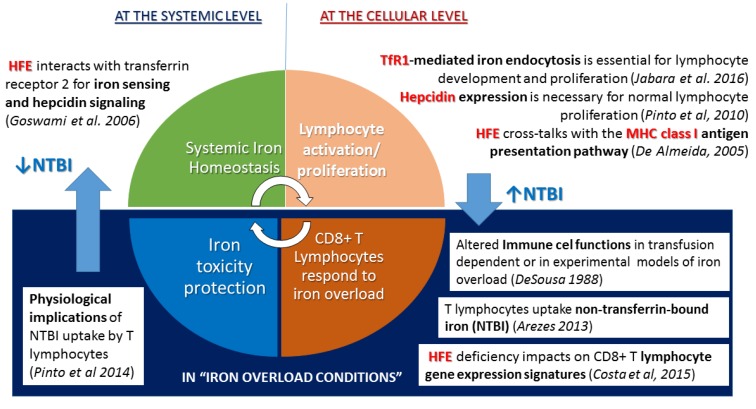
Diagram representation of the dual role of the human hemochromatosis protein HFE as a critical molecule in the reciprocal interactions between iron homeostasis, the major histocompatibility complex (MHC) and lymphocyte functions.

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
