# Peer review of "HFE Related Hemochromatosis: Uncovering the Inextricable Link between Iron Homeostasis and the Immunological System"

_pharmaceuticals, 2019, doi:10.3390/ph12030122_

Round 1

Reviewer 1 Report

The article represents a thorough review of the historical and literature background of the interplay between hemochromatosisenetic predisposition and the immune system. They also present results of a clinical case report focusing on the same issue, followed by a discussion of the future directions and implications of their findings in light of the prvious literature. 

Comment:

1- In the title, "immunological" system, could be replaced by "immune" system.

Author Response

We appreciate the comment and understand that “immune system” is the most generally used term when referring to immunological functions. This term, however, is mostly associated with the classical immune responses against infection or foreign antigens. The reason why we use the more general term “immunological” is to highlight the fact that immunity cannot be seen only as a system to protect against external pathogens but also to the internal threats derived from pathophysiological processes, such is the case of protection against iron toxicity

Reviewer 2 Report

The article titled “HFE RELATED HEMOCHROMOATOSIS: uncovering the inextricable link between iron homeostasis and the immunological system” by Porto and colleagues was reviewed. The article was presented as a “review / hypothesis” design with additional primary “case study” materials discussed for “illustrative” value. Therefore, the merits of this review focus on these aspects of the manuscript and not the rigor and novelty of the presented primary data.

To summarize, this was an exceptionally well written and entertaining read. It is the type of article that non-specialists can read, enjoy and come away from with the feeling that they have learned something broadly relevant. I absolutely endorse publication of this article, however, I do wonder if it is appropriately suited to MDPI: Pharmaceuticals as pharmacotherapeutic options for hemochromatosis are limited to chelation therapy. If the Journal does elect to publish the article, I would recommend the following:

1)      an adequate discussion of the known mechanisms (with pathway diagrams) and perhaps identify drug-targetable molecules in the pathway to help visualize how “iron overload” may modulate immune development.

2)      Adding a discussion pertaining to HH / HFE-modulation of myeloid (monocyte / microglia) characteristics

3)      Figure 1: needs to be a higher resolution image. This version is basically unreadable.

4)      Figure 2: this figure would benefit from having a graphing of “normal values” ± standard error of the mean (s.e.m.)(with shaded ±95% confidence interval) in each of the 3 graphs so that the reader can quickly determine how these subjects deviate from normal population values.

5)      Line 153: switch order of “can iron”; strike “does”.

6)      Line 154: change “progress” to “progresses”

7)      Figure 3 and relevant texts: change use of place holder commas (e.g. 1,5 x 10^6) to decimal points.

8)      Line 169: change TCD4+ and TCD8+ to CD4+ and CD8+ T cells or another standard convention.

9)      Figure 4: change nonT, CD4T and CD8T to be consistent with the text nomenclature

10)   Figure 4: How was total nonT lymph measured (what markers and gating strategy)? If the other blood cell types were measured / measurable, then show the data for each type. You could briefly describe these methods in the Figure legend.

11)   Figure 5: This figure would benefit from a graphing of normal values ± s.e.m. (with shaded ±95% confidence interval) for a convenient reference to the reader

12)   Last sentence, lines 254-256: This is unclear. Please rewrite.

Author Response

To summarize, this was an exceptionally well written and entertaining read. It is the type of article that non-specialists can read, enjoy and come away from with the feeling that they have learned something broadly relevant. I absolutely endorse publication of this article, however, I do wonder if it is appropriately suited to MDPI: Pharmaceuticals as pharmacotherapeutic options for hemochromatosis are limited to chelation therapy. If the Journal does elect to publish the article, I would recommend the following:

1)      an adequate discussion of the known mechanisms (with pathway diagrams) and perhaps identify drug-targetable molecules in the pathway to help visualize how “iron overload” may modulate immune development.

A diagram was included to describe the known pathways in the reciprocal interactions between iron and adaptive immunity and how HFE impacts on those pathways

2)      Adding a discussion pertaining to HH / HFE-modulation of myeloid (monocyte / microglia) characteristics

The suggested discussion was added in the last paragraph of the section “Learning: a brief historical sequence where the improbable led to discovery”

3)      Figure 1: needs to be a higher resolution image. This version is basically unreadable.

A new TIFF version was created for this figure as well as for the other figures

4)      Figure 2: this figure would benefit from having a graphing of “normal values” ± standard error of the mean (s.e.m.)(with shaded ±95% confidence interval) in each of the 3 graphs so that the reader can quickly determine how these subjects deviate from normal population values.

Average and 95% confidence limits (derived from the regressions on age from sex and age matched controls) were added to the graphs.

5)      Line 153: switch order of “can iron”; strike “does”. Changed accordingly.

6)      Line 154: change “progress” to “progresses” Changed accordingly.

7)      Figure 3 and relevant texts: change use of place holder commas (e.g. 1,5 x 10^6) to decimal points.

We changed to decimal points in the text and in figure 1, but the graphs were maintained in their original forms which are derived directly from the statistical program.

8)      Line 169: change TCD4+ and TCD8+ to CD4+ and CD8+ T cells or another standard convention. Changed accordingly

9)      Figure 4: change nonT, CD4T and CD8T to be consistent with the text nomenclature Changed accordingly

 10)   Figure 4: How was total nonT lymph measured (what markers and gating strategy)? If the other blood cell types were measured / measurable, then show the data for each type. You could briefly describe these methods in the Figure legend.

The methods of calculation are now described in the figure legend.

11)   Figure 5: This figure would benefit from a graphing of normal values ± s.e.m. (with shaded ±95% confidence interval) for a convenient reference to the reader

The graphs now include the regression line and the 95% confidence limits. We updated the analysis of the control population that has now a n=190.

12)   Last sentence, lines 254-256: This is unclear. Please rewrite. The sentence was rewritten.